# Predicting Global Terrestrial Biomes with the LeNet Convolutional-Neural-Network

Hisashi SATO[1], Takeshi ISE[2]

[1]Research Institute for Global Change (RIGC), Japan Agency for Marine-Earth Science and Technology (JAMSTEC), Yokohama, 236-0001, JAPAN
[2]Field Science Education and Research Center (FSERC), Kyoto University, Kyoto, 606-8502, JAPAN

*Correspondence to*: Hisashi SATO (hsatoscb@gmail.com)

**Abstract.** A biome is a major regional ecological community characterized by distinctive life forms and principal plants. Many empirical schemes such as the Holdridge Life Zone (HLZ) system have been proposed and implemented to predict the global distribution of terrestrial biomes. Knowledge of physiological climatic limits has been employed to predict biomes, resulting in more precise simulation, however, this requires different sets of physiological limits for different vegetation classification schemes. Here, we demonstrate an accurate and practical method to construct empirical models for biome mapping: A convolutional neural network (CNN) was trained by an observation-based biome map, as well as images depicting air temperature and precipitation. Unlike previous approaches, which require assumption(s) of environmental constrain for each biome, this method automatically extracts non-linear seasonal patterns of climatic variables that are relevant in biome classification. The trained model accurately simulated a global map of current terrestrial biome distribution. Then, the trained model was applied to climate scenarios toward the end of the 21st century, predicting a significant shift in global biome distribution with rapid warming trends. Our results demonstrate that the proposed CNN approach can provide an efficient and objective method to generate preliminary estimations of the impact of climate change on biome distribution. Moreover, we anticipate that our approach could provide a basis for more general implementations to build empirical models of other climate-driven categorical phenomena.

## 1 Introduction

Terrestrial biomes and climate are among the earliest known ecological concerns, and many empirical schemes have been proposed to characterize their relationship (Prentice and Leemans, 1990). One of the best known of these schemes is the Holdridge Life Zone (HLZ) system (Holdridge, 1947), which classifies vegetation distribution using only two independent variables: the annual mean precipitation and the bio-temperature (i.e., mean of above-freezing air temperature). Due to its simplicity, this scheme has been extensively implemented in numerous studies (Emanuel et al., 1985; Henderson-Sellers, 1991; Lugo et al., 1999; Monserud and Leemans, 1992; Prentice, 1990). For example, Elsen et al. (2021) applied historical climatologies and climate projections to the HLZ system for determining potential changes in global life zone distributions under changing climates.

Despite its relative simplicity, the HLZ scheme accounts well for ecophysiological constraints. This scheme is based on bio-temperatures, given that plant productivity becomes negligible at temperatures below 0°C. Furthermore, it employs logarithmic conversions to better depict the relationship between climatic parameters and life zone boundaries, in quantitative recognition of the temperature control of metabolic processes. However, since the HLZ scheme only considers annual climate means, it cannot account for climatic tolerance (e.g., minimum and maximum temperatures) nor the occurrence and extent of drought seasons, both of which substantially affect biome distribution (Prentice et al., 1992).

Efforts have been made to develop biome-mapping schemes that incorporate these environmental constraints. These implementations are considered to have clear physiological bases (Prentice et al., 1992; Woodward and Williams, 1987), and their predictions simulate present-day distributions of vegetation more accurately than the HLZ scheme. However, an important drawback of this type of approach is that it requires absolute physiological limits for each vegetation type or plant functional type (PFT), for which there is still insufficient comprehensive information, as this cannot be estimated from the geographical distribution of the vegetation (Lavorel et al., 2007). Making matters more difficult, researchers do not share the same classification criteria for terrestrial biomes, and the number of vegetation types or PFTs varies widely from five (Henderson-Sellers, 1991) to almost 100 (Box, 1981), depending on the research purpose and the geographical scale studied.

By contrast, empirical approaches like the HLZ scheme do not require detailed physiological data, and thus have the advantage of being easily applicable to any given vegetation classification criteria. Recently, empirical models for biome mapping using various types of environmental data have been developed by employing multinomial logistic regression (Levavasseur et al., 2012; Levavasseur et al., 2013) and machine learning algorithms (Hengl et al., 2018).

A convolutional neural network (CNN) has been successfully adapted for use in species distribution modelling at regional scales (Benkendorf and Hawkins, 2020; Botella et al., 2018); however, it has not been used to develop global biome models. A CNN is an algorithm for machine learning in which a model learns to conduct classification tasks directly from training data. Model training of a CNN is based on finding patterns in the spatial organization of the training data (typically images) that recognizes its classification well. Unlike other conventional algorithms for machine learning, CNN learns directly from training data without a requirement for manual feature extraction.

Indeed, Botella et al. (2018) empirically demonstrated that a CNN model performed better at reconstructing species distributions than the popular species distribution modelling method, MAXENT (Phillips et al., 2006). This higher performance was attributed to CNN's efficient use of spatial patterns in environmental variables, which often control species distribution. MAXENT ignores these spatial patterns. A second explanation for the improved performance is that CNN can treat high order interaction effects between input variables, whereas MAXENT, like the majority of other methods, only represents interactions between environmental variables by the products of variable pairs.

Using a CNN approach, we demonstrate an accurate and practical method to construct empirical models for operational global biome mapping. After evaluating the accuracy of the biome map reconstructed by this method, we applied the trained CNN to climatic scenarios toward the end of the 21st century to demonstrate a possible model's application to predict the shift in the global biome map under changing climate. To the best of our knowledge, this is the first application of CNN to

reconstruct a global biome map. We only employed a small number of climatic variables for input to examine how CNN improves the reconstruction accuracy compared to the classical HLZ scheme.

We follow Ise and Oba (2019) and Ise and Oba (2020) for training CNN with input variables. This method represents climatic conditions using graphical images and employs them as training data for CNN models. To account for seasonal variability, previous correlative climate-vegetation models needed to pre-define representative variables. For example,

Levavasseur et al. (2013) divided each climatic variable into four "seasonal" predictors by averaging data corresponding 3-month periods (i.e., DJF for winter, MAM for spring, JJA for summer, and SON for fall). By contrast, the method we employed can automatically extract non-linear seasonal patterns for climatic variables that are relevant in biome classification. In other words, it enables CNNs to learn the seasonal pattern of multiple climatic variables without any indexical expression, which would reduce the amount of information and add a source of arbitrary.

## 2 Methods

### 2.1 Data

For training the CNN model, we employed potential land cover types and the monthly climate information from the ISLSCP2 Potential Natural Vegetation Cover (Ramankutty and Foley, 2010) and CRU TS4.00 (Harris and Jones, 2017) datasets, respectively. Both datasets have a 0.5° global surface grid resolution. The ISLSCP2 dataset is an observation-based

biome map, which classifies the global land surface into 15 vegetation types (Fig. 1a). The ISLSCP2 dataset represents the world's vegetation cover that would most likely exist now in equilibrium with present-day climate and natural disturbance in the absence of human activities. The CRU TS4.00 is based on an archive of climatic conditions observed in more than 4,000 weather stations distributed worldwide. Climatic conditions between 1971 and 1980 were selected for CNN training since this time period is just before the beginning of a clear global warming trend (Rood, 2015), and the number of meteorological

stations that contributed to the dataset remained relatively stable (Harris et al., 2014).

In machine learning experiments, a fraction of the training data is typically divided randomly into two subsets, of which one is used for model training, and the other is then used to validate the trained model. This study used the CRU TS4.00 climate data as training data, which was generated by interpolating data from weather stations, meaning that values in each grid are not independent of those in nearby grids. Under these circumstances, validation using the typical procedures described above

would risk overfitting (i.e., training the model too closely or exactly to a particular set of data, thereby creating a model that may fail to fit additional data or reliably predict future observations) (Leinweber, 2007). Therefore, other climate datasets were used for validating the trained model: NCEP/NCAR reanalysis (Kalnay et al., 1996), and the HadGEM2-ES (Collins et al., 2011) and MIROC-ESM datasets (Watanabe et al., 2011). Notably, the nature of these three datasets is different from that of the CRU TS4.00; the NCEP/NCAR consists of reanalysis data that incorporates observed and weather model output

data, while the other two datasets were derived only from climate models. Details of these climate datasets are available in

Table S1. To be consistent with the training data, the spatial resolutions of the validation data were linearly interpolated to a 0.5° grid mesh, and climatic conditions from 1971 to 1980 were employed.

In this study, the accuracy when the model was applied to the training climate dataset (i.e., the CRU dataset) is referred to as the "training accuracy", which shows how well the model was trained to extract common features of each category from images. The accuracy for the validation climate dataset (i.e., the NCEP/NCAR reanalysis, Had2GEM-ES, and MIROC-ESM datasets) is referred to as the "test accuracy", which shows how the model is robust against independent input data.

**2.2 Visualization of climate data for machine learning**

We graphically represented the standardized air temperature and precipitation data on a grid using R statistical computing software version 3.3.3 (R-Core-Team, 2018). These images will be referred to hereafter as visualized climatic environments (VCEs). For efficient machine learning, climate data were standardized prior to visualization. The -20–30°C monthly mean air temperature range and 8–400 mm/month precipitation range were log-transformed to 0.01–1.00. Values below and above these ranges were respectively treated as 0.00 and 1.00. To evaluate how seasonality of climate regulates the biome, we also conducted CNN training with annual mean air temperature and annual precipitation. For this analysis, an annual mean bio-temperature range of 0–30°C and an annual precipitation range of 80–4000 mm/year were used. Here, bio-temperature was defined as the mean of above-freezing monthly air temperatures. Using the annual mean bio-temperature and annual precipitation, we first evaluated how different representations of the VCEs influenced the training and found no major differences (Table S2), and hence the most compact VCE with the smallest computation time requirement, the RGB colour tile, was used for this entire study.

In the VCE of the RGB colour tile, up to three climate variables can be represented by RGB channels. To find the optimal combination of climatic variables, we systematically evaluated the model performance of 14 combinations of climatic variable experiments for both annual and monthly means (Tables S3 and S4, respectively). Downward shortwave radiation and humidity were added for this evaluation, as all of the climate datasets contain these. Generally, training accuracy increases with the number of climatic variables; however, the test accuracy does not increase further after two climatic variables. This suggests that models with three climatic variables are at risk of overfitting. Amongst the models of annual and monthly means of climatic variables, the model with monthly mean air temperature and monthly precipitation had the highest test accuracy. Therefore, models that combined air temperature (bio-temperature for the model of annual mean climate) and precipitation were employed for the entire study.

We also evaluated the influences of different transformations of climatic variables (Table S5) and assignment patterns of air temperature and precipitation to RGB colour channels of the VCE (Table S6) on the resulting accuracy. Based on these evaluations, we settled on models with a combination of air temperature (bio-temperature for the model of annual mean climate) and precipitation, both of which are log transformed, and assigned to the blue and red channels, respectively, of the colour tile VCE representation. Examples of VCEs of annual mean climate and monthly mean climate are shown in figures S1 and S2, respectively.

### 2.3 Training of the CNN model

The LeNet (Lecun et al., 1998), which is the world's first CNN, was employed for this study. The computer employed to execute the learning had Ubuntu 16.04 LTS installed as the operating system and was equipped with an Intel Core i7-8700 CPU, 16 GB of RAM, and an NVIDIA GeForce GTX1080Ti graphics card, which accelerates the learning procedure. On the computer, the NVIDIA DIGITS 6.0.0 software (Caffe version: 0.15.14) served as the basis for CNN execution, and LeNet was employed to train the CNN via the TensorFlow library. To see how DIGITS actually implements the CNN, its internal

code can be viewed using the DIGITS menu (on the "New image model" screen, click the "Custom Network tab" and select "TensorFlow"). A description of the CNN model and its parameter settings are available in Supplementary Information 1. To train the CNN model, ten VCEs corresponding to years 1971–1980 were generated for each grid using the CRU data, resulting in 572,640 VCEs (i.e., 10 × 57,264 grids). These VCEs were assigned to 15 categories according to the observation-based biome of the grid, and the CNN model was trained to determine biomes from the VCEs. The numbers of

training VCEs for each biome ranged from 4,490 (comprising temperate broadleaf evergreen forest/woodland areas) to 91,740 (comprising evergreen/deciduous mixed forest area). The training was conducted for each of the annual and monthly sets of VCEs, and their computation times for training completion were 109 and 132 minutes, respectively. The annual and monthly climate training procedures are identical except for its VCEs.

### 2.4 Validation of the trained model

To validate the trained CNN model, a VCE of the average climate conditions from 1971 to 1980 was obtained for each grid and each validation climate dataset. These VCEs were applied to the trained CNN model and were classified by their most plausible biome. It took roughly 8 minutes to complete the VCE classification (i.e., 57,264 in total) for each climate dataset. Then, the computed biome distributions were validated by quantitative comparison with the observation-based biome map of ISLSCP2.

For comparing the differences and similarities between two biome maps, cross-tabulation matrices were obtained for each comparison. Tables S7 and S8 show cross-tabulation matrices of training accuracies as examples. Using these matrices, the differences between the two biome maps were separated into two components: quantity disagreement and allocation disagreement (Pontius and Millones, 2011). Here, a quantity disagreement indicates a discrepancy between the proportions of the categories (i.e., the biome), while an allocation disagreement indicates a discrepancy in the spatial allocation of the

categories under a given set of category proportions in the reference and comparison maps.

The use of one particular climatic dataset for training and three different climatic datasets for validation introduces a source of arbitrary error. To examine the dependency of climatic datasets for training and reconstructing performances, an experiment was performed wherein training and reconstruction of the same biome map was conducted using all combinations of the four historical climatic datasets, and then the reconstructive accuracies were compared.

Ten years of climate data may be insufficient to accurately train the model. We therefore conducted a sensitivity test in which performance was compared among models trained on monthly climate data averaged over 10-year (1971–1980; control), 20-year (1961–1980), and 30-year (1951–1980) periods. Validation datasets for each model were averaged over the same periods as the training data.

We used different climate datasets for training and validating the models to avoid overfitting that may be caused by dependencies in values among nearby grids in the training data (CRU TS4.0). To assess the effects of overfitting, we compared performance among four models that differed with respect to the grain size of training data. Nearby grid cells (0.5°) of the CRU dataset were aggregated by one of four grain sizes: $1 \times 1$ (0.5°), $2 \times 2$ (1.0°), $4 \times 4$ (2.0°), and $8 \times 8$ (4.0°). For each grain size group, 70% of grains were randomly selected for model training, and the remaining were assigned to validation. Validation with coarser grains should be less impacted by overfitting. In addition to the extent of overfitting, grain size may also influence training efficiency, because a coarser grain may skew the allocation ratio of minor biomes between training and validation sub-groups, especially when these biomes have clumped distributions. To assess this possibility, validation was also conducted using other climate datasets.

Finally, we conducted an additional experiment for comparing the accuracy of PNV map reconstruction between the HLZ scheme and our method using common training data set. We developed a look-up table of the most common PNV for each combination of annual mean bio-temperature class and annual precipitation class, consistent with the HLZ scheme. The bin sizes of the HLZ scheme are six for the annual mean bio-temperature class and eight for the annual precipitation class. As these coarse-grained bin-sizes would potentially depress the accuracy of the PNV simulation, we also developed look-up tables of $12 \times 16$, $24 \times 32$, and $48 \times 64$ bin-sizes to ensure that the comparison between our model and the HLZ scheme is as fair as possible. Note that the HLZ scheme employs a hexagon table, but we employed a cross-tabulation table for simplicity. CRU annual climate and ISLSCP2 PNV map were used for generating the table. Then the table was applied to all climatic datasets we employed in this study, drawing reconstructed PNV maps for comparison.

## 2.5 Application of the CNN model to future climate scenarios

Following validation, the CNN model trained with monthly mean climate data was used to predict future biome distribution maps by applying climate scenarios for the 21st century. These predictions were conducted in combinations of two GCMs (i.e., MIROC-ESM and HadGEM2-ES) and two representative concentration pathways (RCPs; i.e., RCP2.6 and RCP8.5). These RCPs represent the atmospheric greenhouse gas (GHG) concentration forecasts adopted by the IPCC for its 5th Assessment Report (AR5) in 2014. RCP2.6 assumes that global annual GHG emissions will peak between 2010 and 2020 and decline substantially afterwards. By contrast, RCP8.5 assumes that emissions will continue to rise throughout the 21st century. The scenarios RCP2.6 and RCP8.5, respectively project that atmospheric CO2 could reach 421 ppm and 936 ppm by the end of the 21st century (IPCC, 2013).

# 3 Results and Discussion

## 3.1 Reconstruction of the current biome distribution with the CNN model

A comparison of the training accuracies between the annual climate model and the monthly climate model demonstrated that simulation of some biomes largely depended on climate seasonality (Figs. 1 and 2). Besides the most plausible biome, the CNN outputs its certainty, which is the probability (in %) of the classification judged by the CNN. Geographical distribution of the certainty clearly showed considering seasonality improves the certainty except Northern parts of South America and African continents where no apparent seasonality exists (Fig. S3). These results are consistent with Prentice et al. (1992), demonstrating that global biome distribution is under substantial controls of climatic tolerance and the occurrence and extent of drought seasons. In fact, seasonality significantly improved the average training accuracies from 3.5% to 61.9% for tropical deciduous forests, 0.4% to 54.8% for temperate broadleaf evergreen forests, and 24.5% to 79.0% for boreal deciduous forests (Tables S7 and S8). The same pattern can be observed in test accuracy comparisons (Figs. 2, 3, and S3), although temperate broadleaf evergreen and boreal deciduous forests were largely absent from Had2GEM-ES and MIROC-ESM, respectively (Fig. 2). These absences would be due to differences in the reconstructed current climate among datasets (Fig. S4). Overall, for all climatic datasets examined, better training and test accuracies were consistently obtained in CNN models trained with monthly mean climate data than in those trained with annual mean climate data (Fig. 4). Thus, the CNN model trained with monthly mean climate data was used for analysis with the climate scenarios in the 21st century.

For all combinations of CNN models and climatic data, the allocation disagreement was much larger than the quantity disagreement: while the allocation disagreement ranged from 0.227 to 0.392, the quantity disagreement varied from 0.037 to 0.200 (Fig. 4). The larger allocation disagreement can be explained by the tendency of observation-based biome distributions to be fragmented over areas with similar climatic conditions (Fig. 1a), while model reconstructed biome distributions had more continuous structures (Figs. 1b, 1c, and 3) (For example, Australian continent). The probability of the most plausible biome tended to be lower for these fragmented regions (Fig. S3), suggesting these regions have climatic conditions suitable for multiple potential biomes. The lower quantity disagreement demonstrated that the CNN model reconstructed the fraction of the global biome composition under the current climatic conditions well. As the main purpose of this research is to develop an empirical model of climatic controls on biome distribution, this would indicate that the reconstructions of biome maps with the CNN models are actually much more accurate for their particular purpose than implied by the accuracies found from the simple map comparison.

Table 1 compares the dependence of reconstruction accuracy on combinations of climate datasets for training and test climate datasets. Accuracies were higher and less variable when the climate dataset for training and testing were identical (0.701–0.734), compared to when these datasets were different (0.394–0.559). These results suggest that uncertainty in historical climate reconstruction and over-fitting are more significant sources of failure in reconstructing biome distribution than the dependency of training on a particular climate dataset.

No major trends were observed in test accuracies in the sensitivity test, which compared performance among models trained using monthly climate averaged over 10-, 20-, and 30-year periods (Table S9). This indicates that climate data averaged over a 10-year period are sufficient for model training. However, long-term climatic conditions are important in controlling biome distribution via extreme climates, which may cause complete reorganization of systems and communities and may provide important opportunities for, and constraints to, plant recruitment. For example, in response to anomalous drought during 2002-2003, regional-scale dieoff of overstory woody plants was observed across southwestern North American woodlands (Breshears, et al., 2005). Considering the effects of extreme climates in the model would be an interesting topic for future study.

Grain size of the training and validation data did not result in noticeable differences in training and test accuracies, with the exception of the CRU dataset (Table S10), demonstrating that the influence of grain size on training efficiency is negligible. In contrast, test accuracies of the CRU dataset were lower at coarser grain sizes, at 80.4%, 78.2%, 76.1%, and 72.2% for the $1 \times 1$, $2 \times 2$, $4 \times 4$, and $8 \times 8$ grain sizes, respectively. These results suggest that dependencies in values among nearby grids in the CRU dataset resulted in overfitting. However, the effect of overfitting appears to have been much smaller than that of systematic differences among climate datasets (Fig. S4); irrespective of grain size, test efficiencies of the CRU dataset were least 19.5% higher than those of other datasets. Therefore, our validation method, which suffers from the systematic differences among climate datasets, should underestimates the actual performance of the models, and performance would be much better than we demonstrated in this manuscript.

Accuracies of PNV reconstructions using the HLZ look-up tables for each climate data-set increase with the resolution of bin sizes for climate classifications (Table 2). It reaches quasi-equilibrium at 24 bio-temperature classes $\times$ 32 precipitation classes, which delivers nearly identical results with the CNN model. This result demonstrates that our VCE method extracts the best possible distribution of the most plausible PNV in a two-dimensional space of climatic variables.

## 3.2 Prediction of biome distribution with the CNN model

The applications of the CNN model to the climate scenarios predicted a significant shift in global biome distributions (Fig. 5) and area coverages (Fig. S5) under rapid warming trends (Figs. S6 and S7). For both GCM outputs, more intense biome shifts were predicted for RCP8.5 than for RCP2.6, but the shift trends remained consistent. The most visible change was the expansion of temperate forests over boreal forests in both North America and Eurasia. Boreal and cold vegetation shrank and its composition changed; tundra areas gave way to boreal forests, while boreal evergreen forests became confined to a narrow strip at higher latitudes. Tropical vegetation remained relatively unchanged, but nearly all tropical deciduous forests in the southern hemisphere were substituted by savanna, which coincided with a reduction in annual precipitation (Figs. S6 and S7).

Given the uncertainty of the climatic predictions derived from the ESMs and RCP scenarios, our analysis of the climate change effect only indicates the potential for considerable changes in biome distribution at the end of the 21st century. Besides, changes in expected biome, which is an equilibrium state of vegetation coverage, are not always accompanied by

immediate changes in actual vegetation. In fact, these time lags can be very long (i.e., decades to millennia) because the adjustment of vegetation to new climate conditions entails a series of plant population dynamics processes, such as seed dispersal, establishment, competition against other existing plants, and reproduction (Sato and Ise, 2012). Even present-day plant species distributions are considered not in equilibrium with present-day climates (e.g., Woodward, 1990). Our study cannot infer such transient changes in vegetation, however, current process base approaches are also not a reliable option for reconstructing plant population dynamic processes at the global scale; biome map predictions under common changing climate scenarios differ significantly from state-of-the-art dynamic global vegetation models (DGVMs) (Pugh et al., 2020). Hence, empirical and top down approaches, like our simulation, should still have an important role to play in approximate mapping of biomes under changing climatic conditions.

### 3.3 Limitations and future directions of our approach

There are two types of approach to mapping biomes: the correlative climate-vegetation approach and process-based approach (Notaro et al., 2012; Yates et al., 2009). We employed the former, which has advantages and disadvantages compared to the latter. An advantage of the correlative approach is that it is relatively straightforward and may be rapidly applied to different climate change scenarios. Indeed, models using the correlative approach are a common tool for predicting the impacts of climate change on biodiversity for conservation planning, because they can be easily used to simultaneously assess large numbers of species (e.g., Thomas et al., 2004).

An important disadvantage of the correlative method is that extrapolating current correlations between climate and biome distributions into the future may lead to seriously biased predictions; strong performance in the present climate does not guarantee similar performance under a new set of climatic conditions that may occur in the future. However, neither Had2GEM-ES (Fig. S3f and Fig. S8a, b) nor MIROC-ESM (Fig. S3h, and Fig. S8c, d) showed apparent expansions of biome uncertainty in projected climatic conditions at the end of the 21st century. This may suggest outside the environmental space of the training data is not conspicuous at the global scale. For quantifying methodological uncertainty might also result from comparing performances between correlative and process-based models in 'unsuitable' outside the environmental space of the training data (Yates et al., 2009).

A second disadvantage of the correlative approach is that it cannot infer impacts of elevated atmospheric $CO_2$ on biome distribution. An increase in $CO_2$ may favour forests over grasslands due to the advantage that C3 plants may gain over C4 plants under such conditions (Bond et al., 2003). Notably, palaeoecological studies have demonstrated that C4 ecosystems were more extensive during the last glacial maximum and decreased in abundance following deglaciation in response to increased atmospheric $CO_2$ concentrations (Ehleringer et al., 1997). Besides, projections of atmospheric $CO_2$ have significant divergence among socioeconomic scenarios from 421 ppm (RCP2.6) to 936 ppm (RCP8.5) at the end of the 21st century.

DGVMs, which use process-based approaches, may facilitate the identification of areas where elevated $CO_2$ may affect biome distribution under projected climates. Indeed, the third phase of the Inter-sectoral Impact Model Inter-comparison

Project, now in progress (Warszawski et al., 2014), includes a sensitivity test for CO2 in which biome distribution is compared between scenarios of both climate and CO2 change, and scenarios of climate change only. We should note, however, that even for current state-of-the-art process-based models, incorporating effects of elevated CO2 is not straightforward due to their complexity; effects appear to be taxon specific, to interact strongly with soil type and climate, and to be highly dependent on nitrogen availability (Korner, 2003; Spinnler et al., 2002).

We must also keep in mind that the correlative climate-vegetation approach ignores feedbacks between vegetation and climate, which are known to influence vegetation distribution at equilibrium (Pitman, 2003). Both Had2GEM-ES and MIROC-ESM explicitly consider climate-vegetation interactions, including dynamic adjustment of biome distribution, and hence its projected climates are the outcomes of such interactions. However, due to the difference in projected distributions of biomes among models, some regions should have mismatched reconstructions of the interactions. Implementing the CNN model with earth system models to dynamically adjust biome distribution to simulated climate distribution would address this issue.

The CNN model was trained with an observation-based biome map, which is composed of natural vegetation only. However, the impact of human activity on ecosystems is now so prevalent, and hence predicting ecosystem changes without explicit consideration of socio-economic systems would be challenging (Ellis, 2015). Therefore, future research might address how current patterns of human activity interact with projected biome changes to reveal regions where these interactive agents align and amplify one another.

This study only considers biome distribution at the 0.5-degree scale. At this scale, climate can be regarded as the dominant factor that determines vegetation composition, and hence correlative climate-vegetation approach fits well in identifying vegetation distribution. However, at more local scales, topography, soil type, and fine-scale biotic and abiotic interactions (e.g., habitat structure, fire, storms) become increasingly important (Willis and Whittaker, 2002). One possible extension of our study is integrating these factors, acting at different spatial scales, into a hierarchical modelling framework (Pearson and Dawson, 2003). Another possible extension is simply adding one more variable that tightly controls PNV at sub-grid scales (such as altitude, slopeness, or slope aspect) into the VCE because one of the three RGB channels is empty in our model. For example, for geographically extrapolating flux data observed at flux tower sites, Gerken et al. (2019) trained artificial neural networks (ANN) using the elevation of each tower site.

Our study adopted the LeNet architecture implementation, which has six hidden layers, to create CNN models. Botella et al. (2018) found that a deep network (six hidden layers) outperformed a shallow network (one hidden layer) for building species distribution models; however, Benkendorf and Hawkins (2020) found that using more than two hidden layers was of no benefit, and argued that the usefulness of deeper networks depends on the size of the training dataset. Therefore, carefully selecting the approximate complexity of architecture implementation may improve model accuracy. We compared performances of models trained by four different types of VCE representation of annual precipitation and average annual bio-temperature, and all models have an almost equal performance (Table S2). This result might indicate that LeNet perfectly extracts at least two variables irrespective of how visualized. Lastly, the default parameters in NVIDIA DIGITS 6.0

remained largely unchanged. Our approach was kept relatively simple to demonstrate the robustness of our concept; however, further improvements to the scheme could be explored by selecting other implementation architectures and systematically testing the effect of parameter modulation.

## 4 Conclusion

Regardless of the limitations discussed above, this study provides an efficient and practical method for generating preliminary estimations of the potentially dramatic impact of climate change on biome distributions. Since this method is simply an application of image classification AI, it demands much less technical skill and computer resources. Reconstruction of global biome distribution substantially improved when climate seasonality was taken into consideration, demonstrating that the method successfully extracted seasonal patterns of climatic variables that are relevant in biome classification. This method could also be applied to building empirical models of other climate-driven phenomena such as cropping systems and the spread of vector-borne diseases, and hence has potential to be a de facto standard for building empirical models across a range of research and application fields.

## Data availability

All data required to reproduce the analyses described herein are publicly available at the following URL/DOI: https://doi.org/10.5281/zenodo.4401233.

## Author contributions

H.S. conceived and conducted the experiments. H.S. and T.I. analysed the results. H.S. wrote the manuscript. H.S. and T.I. reviewed the manuscript.

## Acknowledgements

Anonymous reviewers and Dr. Tobias Gerken provided valuable comments on previous versions of this manuscript. Doctors Shuntaro Watanabe and Yurika Oba of Kyoto University offered technical support regarding issues of deep learning, including the installation of the pertinent computer environments. Dr. Tomohiro Hajima, of the Japan Agency for Marine-Earth Science and Technology, converted the climate data of the MIROC-ESM. Dr. Tomomichi Kato, the topical editor of this manuscript, handled the review process. This work was funded by (1) a Japan Society for the Promotion of Science KAKENHI [Grant Numbers 18H03357 and 17H01477] and (2) the Arctic Challenge for Sustainability II (ArCS II) [Program Grant Number JPMXD1420318865]. The authors declare no conflicts of interest.

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

**Table 1**

CNN model accuracies for biome distribution simulations. These accuracies were obtained using the model trained by the climatic dataset on the row, with the climate dataset on the column as an input reconstruction. Therefore, the shaded cells show the accuracy when the climate datasets for training and reconstruction were identical. For each climate dataset, the monthly mean-temperature and monthly precipitation during 1971 to 1980 were standardized and log transformed, then used for drawing the RGB-colour tile VCEs.

|  | CRU | NCEP/NCAR | Miroc-ESM | HadGEM2-ES |
|---|---|---|---|---|
| CRU | 0.736 | 0.559 | 0.478 | 0.512 |
| NCEP/NCAR | 0.553 | 0.704 | 0.431 | 0.485 |
| Miroc-ESM | 0.540 | 0.394 | 0.701 | 0.417 |
| HadGEM2-ES | 0.430 | 0.505 | 0.450 | 0.712 |

**Table 2**

CNN model and HLZ models accuracies for biome distribution simulations. CNN model corresponds to the top row model of table S2 (a RBG colour tile). Four HLZ models have different bin sizes for climate classifications (bio-temperature class × precipitation class). Each model was trained with the CRU dataset and adapted to the all-climate datasets (i.e., agreements of the CRU dataset correspond to the training accuracy, while other climate data correspond to the test accuracy). In addition, for each climate dataset, the annual mean bio-temperature and annual precipitation from 1971 to 1980 were log-transformed before use.

|  | CNN model | HLZ models | | | |
|---|---|---|---|---|---|
|  |  | 6×8 | 12×16 | 24×32 | 48×64 |
| CRU | 58.3 % | 50.0 % | 54.9 % | 58.1 % | 60.4 % |
| NCEP/NCAR | 45.6 % | 43.2 % | 45.8 % | 44.9 % | 44.0 % |
| Miroc-ESM | 48.6 % | 44.7 % | 46.8 % | 48.2 % | 46.9 % |
| HadGEM2-ES | 41.3 % | 37.2 % | 40.1 % | 40.8 % | 39.5 % |

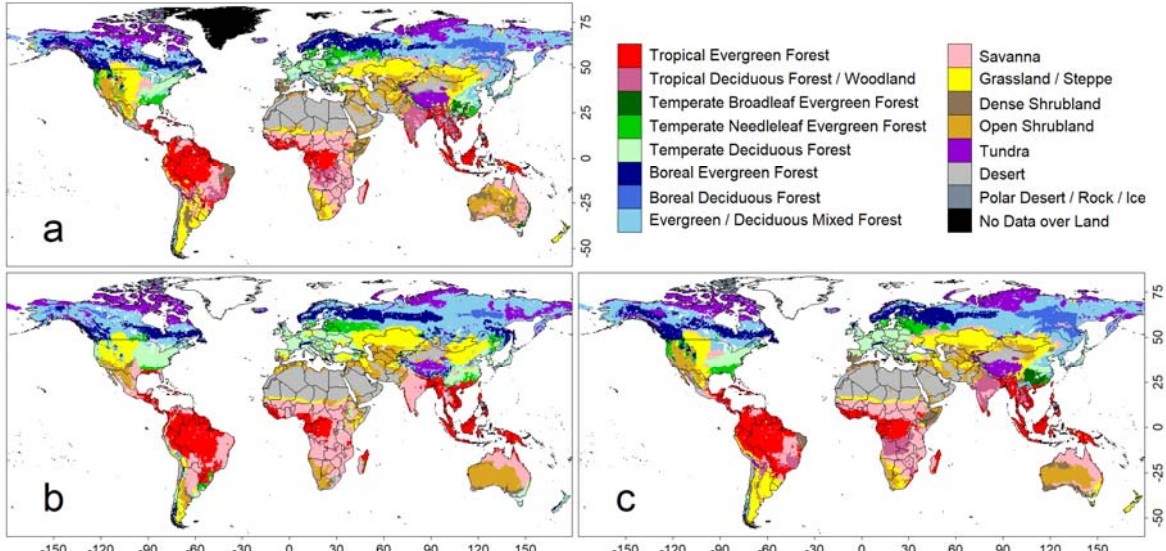

**Figure 1: Comparison of global biome distributions used to evaluate the training accuracies of the convolutional neural network (CNN) model. (a) An observation-based biome map of the ISLSCP2. (b) Biome map derived from the CNN model that was trained with images of annual mean climate. (c) Biome map derived from a CNN model trained with images of monthly mean climate.**

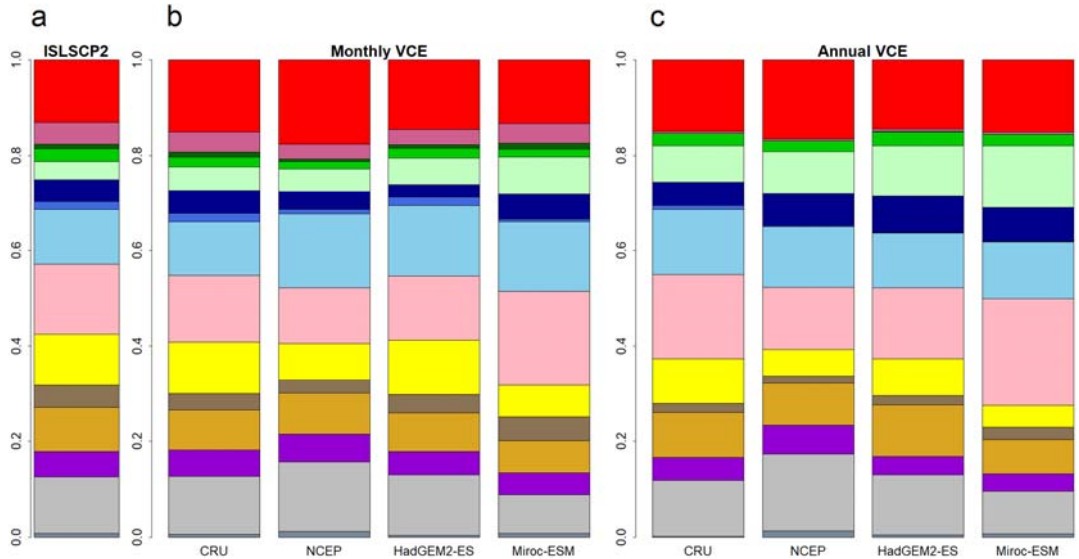

**Figure 2: Global biome compositions of the observation-based map (a) and simulated maps from CNN models trained by monthly mean climate (b) and annual mean climate (c) of CRU climate data spanning from 1971 to 1980. These CNN models were adapted to four climatic datasets (CRU, NCEP, Had2GEM-ES, and MIROC-ESM) spanning the same period of the training data.**

**Figure 3: Test accuracies representing how the trained CNN models simulate a biome map with climatic conditions spanning from 1971 to 1980. (a, c, e) Biome map generated by the CNN model that was trained with annual mean climate images from the CRU dataset. (b, d, f) Biome map generated by the CNN model that was trained by monthly mean climate images from the CRU dataset. Three climatic datasets, which were not involved during the training process, were employed to generate these maps. (a, b) NCEP/NCAR reanalysis data; (c, d) output of the Had2GEM-ES dataset; and (e, f) output of the MIROC-ESM dataset. Colour definitions are available in Figure 1.**

495

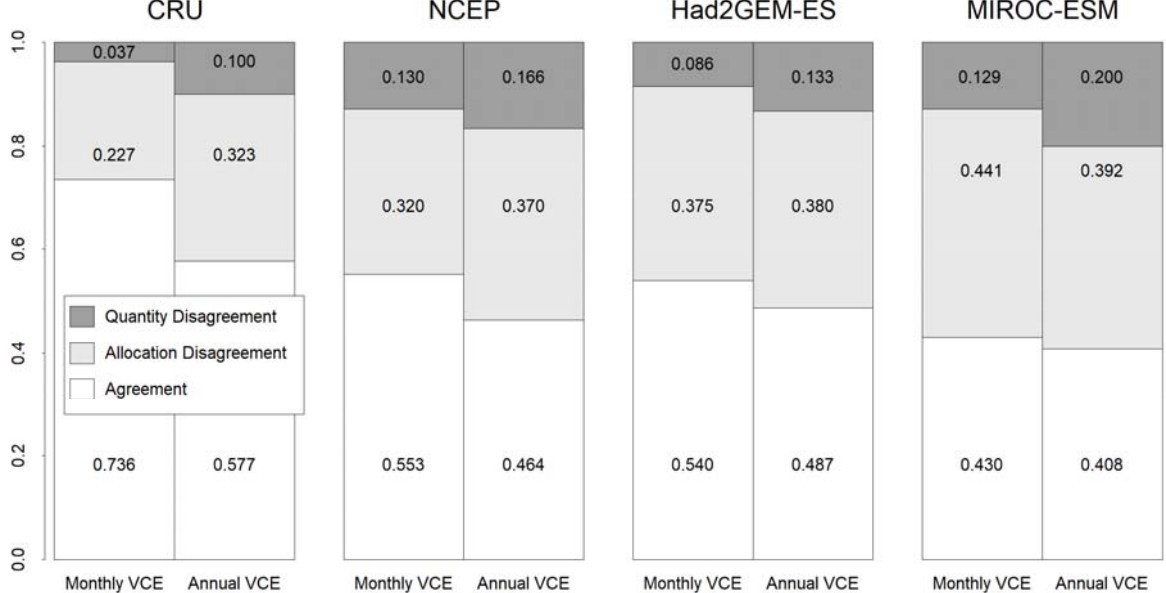

**Figure 4: Fractions of agreement and disagreement between observation-based biome map and simulated biome maps trained by monthly mean climate or annual mean climate from CRU climate data spanning from 1971 to 1980. These CNN models were adapted to one of the four climatic datasets (CRU, NCEP, Had2GEM-ES, and MIROC-ESM) spanning the same period of the training data. The fraction of agreement of the CRU corresponds to the training accuracy, while that of other climate data corresponds to the test accuracy.**

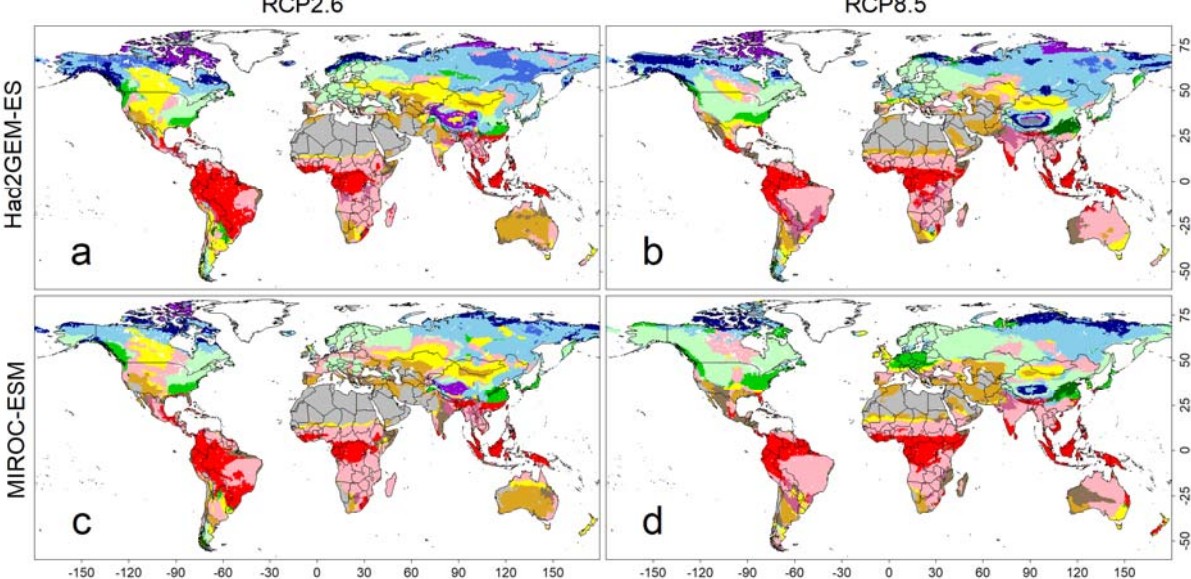

**Figure 5: Predicted biome maps under climatic scenarios from 2091 to 2100. Monthly means of four sets of forecasted climatic conditions derived from combinations of two climate models (i.e., Had2GEM-ES and MIROC-ESM) and two RCP scenarios (i.e., RCP2.6 and RCP8.5). These means were applied to the CNN model that was trained by the current biome distribution map, as well as the present climatic condition derived from the CRU dataset. Colour definitions are available in Figure 1.**

510