# Peer review of "Predicting Global Terrestrial Biomes with the LeNet Convolutional-Neural-Network"

_Geoscientific Model Development, 2021_

## Author Comment (AC2)

Dear Referee #1

Thank you for taking the time to evaluate our manuscript. We feel your comments are pretty constructive for increasing the quality of the expected revised manuscript. Following is our one-to-one response to your concerns. Throughout this letter, your comments are written in blue color and are numbered.

(1) In this manuscript, the authors applied a data-driven or machine learning approach, the conventional neural network (CCN), to estimate global distribution of the potential vegetation types. They evaluated accuracy of retrieving the present state, and then estimated future shifts under projected climate change. Finally, they discussed the merits and limitations of the empirical approach.

**Response:**

Yes, it is an accurate abstract of our work.

(2) My first impression of the manuscript is that this is a mixture of old problem and new technique. Such revisiting is sometimes effective, but only if the new technique provides deeper insights and/or apparently higher comprehensiveness than those in precedented studies. In my view, regretfully, I could not find enough advancements in this study; it looks like an exercise of the CCN.

**Response:**

Our approach has higher comprehensiveness than previous studies: it automatically extracts non-linear seasonal patterns for climatic variables relevant to biome classification.

The Holdridge Life Zone only considers annual climate means, and hence it cannot account for seasonal patterns of climatic condition, which affect biome distribution. Accordingly, many subsequence studies of biome mapping tried to incorporate seasonal patterns by assuming environmental constraints (such as tolerance of drought) for each plant group (such as plant functional types, biomes, or vegetation types). These approaches require absolute physiological limits for each plant group. However, there is no straightforward way to estimate such limits because plant groups contain a large number of species. By taking advantage of CNN, our approach can provide an easy, efficient, accurate, and comprehensive solution for this issue. To clarify and emphasize this issue, we will insert the following sentence in the expected revised manuscript's abstract (line 14).

-- Unlike previous approaches, which require assumption(s) of environmental constrain for each biome, this method automatically extracts non-linear seasonal patterns of climatic variables that are relevant in biome classification.

Also, we will insert the following sentence in conclusion (line 283)

-- Reconstruction of global biome distribution substantially improved when climate seasonality was taken into consideration, demonstrating that the method successfully extracted seasonal patterns of climatic variables that are relevant in biome classification.

(3) In other words, I am unsure whether this manuscript falls within the scope of Geoscientific Model Development.

**Response:**

The authors' instruction of the Geoscientific Model Development (https://www.geoscientific-model-development.net/) defines scopes of manuscript types for considering peer-reviewed publication. Our manuscript satisfies the following items, and hence we are sure this manuscript falls within the scopes of Geoscientific Model Development.

Geoscientific model descriptions, from statistical models to box models to GCMsModel experiment descriptions, including experimental details and project protocols

(4) The manuscript is short and well-focused but need more methodological descriptions and insightful discussion.

**Response:**

Description concerning machine specificity and software environment will be moved from supplementary information 1 (lines 23-26) to the main body (line 122). To keep the main body short, we hope to stay parameter setting descriptions at the supplementary information 1. Concerning discussion, we are happy to add more text if you specify what is not enough.

(5) The manuscript starts from several statements about the Holdridge Life Zone, but I think this part is unnecessary.

**Response:**

As you mentioned, there are several statements about the Holdridge Life Zone in the introduction. As our approach is a kind of an extension of The Holdridge Life Zone, we hope these statements to be maintained.

(6) On the other hand, the authors gave few words on remote sensing of vegetation, even for validation of the estimation result.

**Response:**

Please refer our response on the item (9).

(7) As the authors discussed, the data-driven approach has limitations. The model may not be applicable to the states outside the range of trained data, and the present CCN model used only temperature and precipitation as input data. Namely, it did not account for the effects of atmospheric CO2, nutrient, and disturbance, each of which is hot issues in the study area and so needs further discussions. I agree with the meaning of examining the potential vegetation, because natural disturbances and human impacts (e.g., land-use) are too complicated to discuss climatic impacts on global-scale vegetation. In this regard, the study is one of a few attempts to apply the machinelearning method to capture the potential vegetation. However, becaus of critical limitations and deficiencies, I cannot recommend accepting the manuscript for publication.

**Response:**

As you pointed out, our approach ignores atmospheric CO2, nutrients, and disturbances like other equilibrium and niche models. Besides, it also ignores other mechanisms that can impact real-world responses and vegetative state transitions (such as reproduction times, dispersal abilities/limitations, and geographical barriers to migration). Nevertheless, our approach quickly assesses the degree to which potential natural vegetation (PNV) states are projected to persist or shift under climate change globally. Our approach provides one of the few applications of CNN at a global assessment of spatiotemporal dynamics among PNV using standardized, empirical, and ecologically relevant climate information.

Indeed, after submitting our manuscript, Elsen *et al.* (2021) published an article where they adapted the Holdridge life zone for evaluating how changing climate shifts terrestrial life zone. Our approach has a clear advantage to the study in considering seasonal patterns of the climatic condition by applying CNN. Elsen, P. R., et al. (2021). "Accelerated shifts in terrestrial life zones under rapid climate change." Global Change Biology.

Your criticism is reasonable of cause, but it describes general limitations of whole studies employing the so-called "climatic envelope approach" not specific to our particular study. Besides, process-based approaches are also unreliable options, as explained in our manuscript (line 228-233, line 254-260). At this moment, it cannot be said which is the better approach for projecting global PNV distribution under changing climate.

For showing an example that a climatic envelope approach is used as a vital option for projecting biome map, we will add the following sentence, which refers to a recent study (at line 26).

-- For example, Elsen et al. (2021) applied historical climatologies and climate projections to the HLZ system for determining potential changes in global life zone distributions under changing climates.

**Minor points**

(8) Introduction: As mentioned in my general comments, Introduction starts from classic studies. I recommend putting more focus on modern and recent studies.

**Response:**

Please refer our response on the item (7).

(9) Line 72: I am quite unsure why the ISLSCP2 data were selected as benchmark data of potential vegetation and why any remote sensing data were not used.

**Response:**

Following is a part of the description of the ISLSCP II Potential Natural Vegetation Cover dataset (https://daac.ornl.gov/cgi-bin/dsviewer.pl?ds\_id=961), and it addresses your concern.

"The geographic distribution of contemporary land cover types can be derived from remotely-sensed data. However, humans now dominate much of the world and there is little evidence of the pre-human-settlement natural vegetation or Potential Natural Vegetation (PNV). PNV, as defined here, does not necessarily represent the world's natural pre-human-disturbance vegetation. Rather, our definition of PNV represents the world's vegetation cover that would most likely exist now in equilibrium with present-day climate and natural disturbance, in the absence of human activities."

To clarify the nature of the dataset, and to clarify the aim of our study, we will insert the following sentences in line 74.

-- The ISLSCP2 dataset represents the world's vegetation cover that would most likely exist now in equilibrium with present-day climate and natural disturbance in the absence of human activities.

Besides, the "ISLSCP2" in line72 will be replaced by "ISLSCP2 Potential Natural Vegetation Cover" because ISLSCP2 is just the name of a project name, not a name of a dataset.

(10) Line 84: Please give references to NCEP/NCAR, HadGEM2-ES, and MIROC-ESM.

**Response**:**

We will!

(11) Line 102: Did you used daily temperature? Or, monthly?

**Response:**

It's monthly. Thanks for finding our missing description!

**(12) Line 129: The computational times should depend on machine ability.**

**Response:**

Right, it primarily depends on a graphics card. The following description, which explained machine specificity, will be moved from supplementary information 1 (lines 23-26) to the main body (line 122). Information about computational time would be helpful for readers as it provides rough estimates of computation cost.

"The computer employed to execute the learning had Ubuntu 16.04 LTS installed as the operating system and was equipped with an Intel Core i7-8700 CPU, 16 GB of RAM, and an NVIDIA GeForce GTX1080Ti graphics card, which accelerates the learning procedure. On the computer, the NVIDIA DIGITS 6.0.0 software (Caffe version: 0.15.14) served as the basis for CNN execution, and LeNet was employed to train the CNN via the TensorFlow library."

(13) Line 172: I could not find explanation about the "certainty" of the CCN output in the method sections.

**Response:**

We will insert the following phrase (Line 172).

-- ", which is the probability (in %) of the classification judged by the CNN."

**(14) Line 190: "quantity" may be removed.**

**Response:**

In this section, "allocation disagreement" and "quantity disagreement" are distinguished. Your mentioned sentence describes quantity disagreement, so we cannot remove "quantity."

(15) Line 195: Table S9 should be moved to main body. Otherwise, you may rewrite this sentence.

**Response:**

As you suggested, Table S9 will be moved to the main body as Table 1. With this change, Tables S10 and S11 in the previous manuscript will be renumbered as tables S9 and S10, respectively.

(16) Line 203: Did you mean stand-replacing disturbances such as wildfire and wind throw? It may be better to provide several examples.

**Response:**

Here, we intended anomalous climate events that have catastrophic influences on plant mortality, and hence we will insert the following sentence in line 204.

-- For example, in response to anomalous drought during 2002-2003, regional-scale dieoff of overstory woody plants was observed across southwestern North American woodlands (Breshears, *et al.*, 2005).

(17) Line 213: I could not understand the sentence. Why the model should have better performance than you showed, if the CRU dataset had high efficiency irrespective of grain size?

**Response:**

We will replace the mentioned sentence as follows.

Previous: "Therefore, our validation method underestimates the actual performance of the models, and performance is much better than we demonstrated in this

**manuscript."**

New: "Therefore, our validation method, which suffers from the systematic differences among climate datasets, should underestimate the actual performance of the models, and performance would be much better than we demonstrated in this manuscript.

(18) Line 249-253: Here, you mentioned about the limitation associated with the atmospheric CO2 concentration. Indeed, atmospheric CO2 levels in 2100 are largely different between RCP2.6 and RCP8.5. So, you should make more discussions about this limitation.

**Response:**

We will insert the following phrases at the end of line 253.

-- Besides, projections of atmospheric CO2 have significant divergence among socioeconomic scenarios from 421 ppm (RCP2.6) to 936 ppm (RCP8.5) at the end of the 21st century.

(19) Line 275: At this very last part of the manuscript, you first mentioned about the hardware issue (NVIDIA DIGITS 6.0).

**Response:**

For introducing the NVIDIA DIGITS 6.0 before this sentence, the following description in supplementary information 1 (lines 23-26) will be moved to the main text (line 122).

"The computer employed to execute the learning had Ubuntu 16.04 LTS installed as the operating system and was equipped with an Intel Core i7-8700 CPU, 16 GB of RAM, and an NVIDIA GeForce GTX1080Ti graphics card, which accelerates the learning procedure. On the computer, the NVIDIA DIGITS 6.0.0 software (Caffe version: 0.15.14) served as the basis for CNN execution, and LeNet was employed to train the CNN via the TensorFlow library."

Best, Hisashi SATO (on the behalf of all co-authors)

---

## Author Comment (AC3)

Dear Referee #2

Thank you for taking the time to evaluate our manuscript. Your comments will lead to a thorough revision of the paper.

Following is our one-to-one response to your concerns. Throughout this letter, your comments are written in blue color and are numbered.

(1) This paper presents research into predicting global terrestrial biomes with a CNN using a correlative climate-vegetation approach. The manuscript is of very high standard w.r.t. introducing the problem and motivation, describing the approach, discussing the results and pointing out the limitations. The authors also correctly state that the presented study is not a ground breaking new innovation, but a demonstration of how existing tools can be used in the context of predicting the future of complex systems.

**Response:**

We greatly appreciate your positive evaluation for our manuscript!

(2) Besides the limitations discussed in section 3.3, a few additional aspects come to mind. Firstly, most climate models have no dynamic vegetation models built in. In addition to what the authors stated regarding the lack of feedback between vegetation and climate, it is also known that large ecosystems create their own climate and therefore changes to the ecosystems - due to whatever factor - may affect the future climate as well.

**Response:**

For addressing this issue, the phrases in lines 261-263 will be divided into two parts and rewritten. The original paragraph will be replaced by the new phrases 1. The new phrases 2 will be inserted into line 228.

Previous phrases (Lines 261-263):

We must also keep in mind that the correlative climate-vegetation approach ignores feedbacks between vegetation and climate, which are known to influence vegetation distribution at equilibrium (Pitman, 2003), as well as present-day species distributions that are not in equilibrium with present-day climates (e.g., Woodward, 1990).

New phrases 1 (will be replaced with the previous phrases):

We must also keep in mind that the correlative climate-vegetation approach ignores feedbacks between vegetation and climate, which are known to influence vegetation distribution at equilibrium (Pitman, 2003). Both Had2GEM-ES and MIROC-ESM explicitly consider climate-vegetation interactions, including dynamic adjustment of biome distribution, and hence its projected climates are the outcomes of such interactions. However, due to the difference in projected distributions of biomes among models, some regions should have mismatched reconstructions of the interactions. Implementing the CNN model with earth system models to dynamically adjust biome distribution to simulated climate distribution would address this issue.

New phrases 2 (will be inserted into line 228):
Even present-day plant species distributions are considered not in equilibrium with present-day climates (e.g., Woodward, 1990).

(3) It is also not clear to me how to separate the human effects that are partly, implicitly included in the model (e.g. human-made landuse changes in the training period) and, more important, the ones that are not included. Recent and future rapid development, sealing of surfaces, large-scale deforestation and irrigation, large-scale relocation of humans due to rising sea levels and temperatures, the development and use of genetic manipulated crops etc. are all factors that may influence future terrestrial biomes. It would be nice to see section 3.3 expanded to include some of this in the discussion and, if possible, to include some suggestions on how to incorporate these complex interactions in a next step.

**Response:**
We will add the following new paragraph at the end of line 263.

-- The CNN model was trained with an observation-based biome map, which is composed of natural vegetation only. However, the impact of human activity on ecosystems is now so prevalent, and hence predicting ecosystem changes without explicit consideration of socio-economic systems would be challenging (Ellis, 2015). Therefore, future research might address how current patterns of human activity interact with projected biome changes to reveal regions where these interactive agents align and amplify one another.

Ellis, E. C. (2015). Ecology in an anthropogenic biosphere. Ecological Monographs, 85, 287-331. https://doi.org/10.1890/14-2274.1

(4) Lastly, there seem to be a bit of a mixup of present and past tense in section 3.1 that should be made consistent. For example, Line 189-190, "The probability of the most plausible biome tend to be ..." (where it should be tends if it is present tense) versus line 184, "... the allocation disagreement was much larger ...".

**Response:**

Thanks for pointing out the mixup of present and past tense. We check the throughout "Results and Discussion" part. In the expected revised manuscript, all phrases describing results will be uniformed to past tense, while all terms concerning discussion will be uniformed to present tense. Specific changes will be as follows.

Line 172: "shows"      will be "showed"
Line 187: "have"       will be "had"
Line 189: "tend"       will be "tended"
Line 195: "compared" will be "compares"
Line 244: "shows"      will be "showed"

Best,
Hisashi SATO (on the behalf of all co-authors)

---

## Author Response (AR1)

**Dear editor,**

We want to thank the referees for their constructive comments, and we appreciate your help and patience. We have revised the manuscript according to the referee's comments and suggestions. Below is our one-to-one response to your concerns. Throughout this letter, given words are written in blue and numbered consecutively.

Kind regards, Hisashi SATO on behalf of all authors

**Referee #1**

(1) In this manuscript, the authors applied a data-driven or machine learning approach, the conventional neural network (CCN), to estimate global distribution of the potential vegetation types. They evaluated accuracy of retrieving the present state, and then estimated future shifts under projected climate change. Finally, they discussed the merits and limitations of the empirical approach.

**Response:**

Yes, it is an accurate abstract of our work.

(2) My first impression of the manuscript is that this is a mixture of old problem and new technique. Such revisiting is sometimes effective, but only if the new technique provides deeper insights and/or apparently higher comprehensiveness than those in precedented studies. In my view, regretfully, I could not find enough advancements in this study; it looks like an exercise of the CCN.

**Response:**

Our approach has higher comprehensiveness than previous studies: it automatically extracts non-linear seasonal patterns for climatic variables relevant to biome classification.

The Holdridge Life Zone only considers annual climate means, and hence it cannot account for seasonal patterns of climatic condition, which affect biome distribution. Accordingly, many subsequence studies of biome mapping tried to incorporate seasonal patterns by assuming environmental constraints (such as tolerance of drought) for each plant group (such as plant functional types, biomes, or vegetation types). These approaches require absolute physiological limits for each plant group. However, there is no straightforward way to estimate such limits because plant groups contain a large number of species. By taking advantage of CNN, our approach can provide an easy, efficient, accurate, and comprehensive solution for this issue.

To clarify and emphasize this issue, we inserted the following sentence in the revised manuscript's abstract (L14).

-- Unlike previous approaches, which require assumption(s) of environmental constrain for each biome, this method automatically extracts non-linear seasonal patterns of climatic variables that are relevant in biome classification.

Also, we inserted the following sentence in conclusion (L327)

-- Reconstruction of global biome distribution substantially improved when climate seasonality was taken into consideration, demonstrating that the method successfully extracted seasonal patterns of climatic variables that are relevant in biome classification.

(3) In other words, I am unsure whether this manuscript falls within the scope of Geoscientific Model Development.

**Response:**

The authors' instruction of the Geoscientific Model Development (https://www.geoscientific-model-development.net/) defines scopes of manuscript types for considering peer-reviewed publication. Our manuscript satisfies the following items, and hence we are sure this manuscript falls within the scopes of Geoscientific Model Development.

-- Geoscientific model descriptions, from statistical models to box models to GCMs

-- Model experiment descriptions, including experimental details and project protocols

(4) The manuscript is short and well-focused but need more methodological descriptions and insightful discussion.

**Response:**

Descriptions concerning machine specificity and software environment were moved from supplementary information 1 to the main body (L128-134). To keep the main body short, we hope to stay parameter setting descriptions at the supplementary information 1. Concerning discussion, we are happy to add more words if you specify what is not enough; however, we added an additional experiment and debate in this revision (please refer our responses on items 21, 22, 24, 26, and 30 below). We hope these changes made the discussion more insightful.

(5) The manuscript starts from several statements about the Holdridge Life Zone, but I think this part is unnecessary.

**Response:**

As you mentioned, there are several statements about the Holdridge Life Zone in the introduction. As our approach is a kind of an extension of The Holdridge Life Zone, we hope these statements to be maintained.

(6) On the other hand, the authors gave few words on remote sensing of vegetation, even for validation of the estimation result.

**Response:**

Please refer our response on the item (9).

(7) As the authors discussed, the data-driven approach has limitations. The model may not be applicable to the states outside the range of trained data, and the present CCN model used only temperature and precipitation as input data. Namely, it did not account for the effects of atmospheric CO2, nutrient, and disturbance, each of which is hot issues in the study area and so needs further discussions. I agree with the meaning of examining the potential vegetation, because natural disturbances and human impacts (e.g., land-use) are too complicated to discuss climatic impacts on global-scale vegetation. In this regard, the study is one of a few attempts to apply the machinelearning method to capture the potential vegetation. However, becaus of critical limitations and deficiencies, I cannot recommend accepting the manuscript for publication.

**Response:**

As you pointed out, our approach ignores atmospheric CO2, nutrients, and disturbances like other equilibrium and niche models. Besides, it also ignores other mechanisms that can impact real-world responses and vegetative state transitions (such as reproduction times, dispersal abilities/limitations, and geographical barriers to migration). Nevertheless, our approach quickly assesses the degree to which potential natural vegetation (PNV) states are projected to persist or shift under climate change globally. Our approach provides one of the few applications of CNN at an assessment of spatiotemporal dynamics among PNV using standardized, empirical, and ecologically relevant climate information.

Indeed, after submitting our manuscript, Elsen *et al.* (2021) published an article where they adapted the Holdridge life zone for evaluating how changing climate shifts terrestrial life zone. Our approach has a clear advantage to the study in considering seasonal patterns of the climatic condition by applying CNN.

Elsen, P. R., et al. (2021). "Accelerated shifts in terrestrial life zones under rapid climate change." Global Change Biology.

Your criticism is reasonable of cause, but it infers general limitations of whole studies employing the so-called "climatic envelope approach" not specific to our particular study. Besides, process-based approaches are also unreliable options, as explained in our manuscript (L258-263, L 286-292). At this moment, it cannot be judged which is the better approach for projecting global PNV distribution under changing climate.

For showing an example that a climatic envelope approach is used as a vital option for projecting biome map, we added the following sentence, which refers to a recent study (L28).

-- For example, Elsen et al. (2021) applied historical climatologies and climate projections to the HLZ system for determining potential changes in global life zone distributions under changing climates.

**Minor points**

(8) Introduction: As mentioned in my general comments, Introduction starts from classic studies. I recommend putting more focus on modern and recent studies.

**Response:**

Please refer our response on the items (5) and (7).

(9) Line 72: I am quite unsure why the ISLSCP2 data were selected as benchmark data of potential vegetation and why any remote sensing data were not used.

**Response:**

Following is a part of the description of the ISLSCP II Potential Natural Vegetation Cover dataset (https://daac.ornl.gov/cgi-bin/dsviewer.pl?ds\_id=961), and it would address your concern.

"The geographic distribution of contemporary land cover types can be derived from

remotely-sensed data. However, humans now dominate much of the world and there is little evidence of the pre-human-settlement natural vegetation or Potential Natural Vegetation (PNV). PNV, as defined here, does not necessarily represent the world's natural pre-human-disturbance vegetation. Rather, our definition of PNV represents the world's vegetation cover that would most likely exist now in equilibrium with present-day climate and natural disturbance, in the absence of human activities."

To clarify the nature of the dataset, and to clarify the aim of our study, we inserted the following sentence in L78.

-- The ISLSCP2 dataset represents the world's vegetation cover that would most likely exist now in equilibrium with present-day climate and natural disturbance in the absence of human activities.

Besides, at the first use of the word "ISLSCP2" (L76), it was supplemented as "ISLSCP2 Potential Natural Vegetation Cover" because ISLSCP2 is just the name of a project name, not a name of a dataset.

**(10) Line 84: Please give references to NCEP/NCAR, HadGEM2-ES, and MIROC-ESM.**

**Response:**

We did! (L90-91)

**(11) Line 102: Did you used daily temperature? Or, monthly?**

**Response:**

It's monthly. We corrected it (L108). Thanks for finding our missing description!

(12) Line 129: The computational times should depend on machine ability.

**Response:**

Right, it primarily depends on a graphics card. The following description, which explained machine specificity, was moved from supplementary information 1 to the main body (L128). Information about computational time would be helpful for readers as it provides rough estimates of computation cost.

"The computer employed to execute the learning had Ubuntu 16.04 LTS installed as the operating system and was equipped with an Intel Core i7-8700 CPU, 16 GB of RAM, and an NVIDIA GeForce GTX1080Ti graphics card, which accelerates the learning procedure. On the computer, the NVIDIA DIGITS 6.0.0 software (Caffe version: 0.15.14) served as the basis for CNN execution, and LeNet was employed to train the CNN via the TensorFlow library."

(13) Line 172: I could not find explanation about the "certainty" of the CCN output in the method sections.

**Response:**

We inserted the following phrase (L190).

-- which is the probability (in %) of the classification judged by the CNN.

**(14) Line 190: "quantity" may be removed.**

**Response:**

In this section, "allocation disagreement" and "quantity disagreement" are distinguished. Your mentioned sentence (L208-209) describes quantity disagreement, so we cannot remove "quantity."

(15) Line 195: Table S9 should be moved to main body. Otherwise, you may rewrite this sentence.

**Response:**

As you suggested, Table S9 was moved to the main body as Table 1. With this change, Tables S10 and S11 in the previous manuscript were renumbered as tables S9 and S10, respectively.

(16) Line 203: Did you mean stand-replacing disturbances such as wildfire and wind throw? It may be better to provide several examples.

**Response:**

Here, we intended anomalous climate events that have catastrophic influences on plant mortality, and hence we inserted the following sentence in L222.

-- For example, in response to anomalous drought during 2002-2003, regional-scale dieoff of overstory woody plants was observed across southwestern North American woodlands (Breshears, *et al.*, 2005).

(17) Line 213: I could not understand the sentence. Why the model should have better performance than you showed, if the CRU dataset had high efficiency irrespective of

**grain size?**

**Response:**

We replaced the mentioned sentence as follows.

Previous: "Therefore, our validation method underestimates the actual performance of the models, and performance is much better than we demonstrated in this manuscript."

New (L232): "Therefore, our validation method, which suffers from the systematic differences among climate datasets, should underestimate the actual performance of the models, and performance would be much better than we demonstrated in this manuscript.

(18) Line 249-253: Here, you mentioned about the limitation associated with the atmospheric CO2 concentration. Indeed, atmospheric CO2 levels in 2100 are largely different between RCP2.6 and RCP8.5. So, you should make more discussions about this limitation.

**Response:**

We appended the following phrases immediately after your mentioned sentence. (L283)

-- Besides, projections of atmospheric CO2 have significant divergence among socioeconomic scenarios from 421 ppm (RCP2.6) to 936 ppm (RCP8.5) at the end of the 21st century.

(19) Line 275: At this very last part of the manuscript, you first mentioned about the hardware issue (NVIDIA DIGITS 6.0).

**Response:**

By the modification according to the item (12) above, NVIDIA DIGITS 6.0 was introduced before this sentence in the revised manuscript.

**Referee #2**

(20) This paper presents research into predicting global terrestrial biomes with a CNN using a correlative climate-vegetation approach. The manuscript is of very high standard w.r.t. introducing the problem and motivation, describing the approach, discussing the results and pointing out the limitations. The authors also correctly state that the presented study is not a ground breaking new innovation, but a demonstration of how existing tools can be used in the context of predicting the future of complex systems.

**Response:**

We greatly appreciate your positive evaluation for our manuscript!

(21) Besides the limitations discussed in section 3.3, a few additional aspects come to mind. Firstly, most climate models have no dynamic vegetation models built in. In addition to what the authors stated regarding the lack of feedback between vegetation and climate, it is also known that large ecosystems create their own climate and therefore changes to the ecosystems - due to whatever factor - may affect the future climate as well.

**Response:**

For addressing this issue, the sentence in lines 261-263 was divided into two parts and rewritten.

**Previous sentence (Lines 261-263):**

-- We must also keep in mind that the correlative climate-vegetation approach ignores feedbacks between vegetation and climate, which are known to influence vegetation distribution at equilibrium (Pitman, 2003), as well as present-day species distributions that are not in equilibrium with present-day climates (e.g., Woodward, 1990).

**New paragraph (inserted in L293):**

-- We must also keep in mind that the correlative climate-vegetation approach ignores feedbacks between vegetation and climate, which are known to influence vegetation distribution at equilibrium (Pitman, 2003). Both Had2GEM-ES and MIROC-ESM explicitly consider climate-vegetation interactions, including dynamic adjustment of biome distribution, and hence its projected climates are the outcomes of such interactions. However, due to the difference in projected distributions of biomes among models, some regions should have mismatched reconstructions of the interactions. Implementing the CNN model with earth system models to dynamically adjust biome distribution to simulated climate distribution would address this issue.

**New sentence (inserted into L257):**

-- Even present-day plant species distributions are considered not in equilibrium with present-day climates (e.g., Woodward, 1990).

(22) It is also not clear to me how to separate the human effects that are partly, implicitly included in the model (e.g. human-made landuse changes in the training period) and, more important, the ones that are not included. Recent and future rapid development, sealing of surfaces, large-scale deforestation and irrigation, large-scale relocation of humans due to rising sea levels and temperatures, the development and use of genetic manipulated crops etc. are all factors that may influence future terrestrial biomes. It would be nice to see section 3.3 expanded to include some of this in the discussion and, if possible, to include some suggestions on how to incorporate these complex interactions in a next step.

**Response:**

We inserted and additional discussion about human-land-use (L300).

-- The CNN model was trained with an observation-based biome map, which is composed of natural vegetation only. However, the impact of human activity on ecosystems is now so prevalent, and hence predicting ecosystem changes without explicit consideration of socio-economic systems would be challenging (Ellis, 2015). Therefore, future research might address how current patterns of human activity interact with projected biome changes to reveal regions where these interactive agents align and amplify one another.

(23) Lastly, there seem to be a bit of a mixup of present and past tense in section 3.1 that should be made consistent. For example, Line 189-190, "The probability of the most plausible biome tend to be ..." (where it should be tends if it is present tense) versus line 184, "... the allocation disagreement was much larger ...".

**Response:**

Thanks for pointing out the mixup of present and past tense. We check the throughout "Results and Discussion" part. In the revised manuscript, all phrases describing results were uniformed to past tense, while all terms concerning discussion were uniformed to present tense.

**Referee #3**

(24) Scope: The manuscript applies and tests the LeNET CNN, which is a published and widely applied CNN to predicting biomes, which is a new application to for this

particular CNN. I am not sure to what extent such an experiment falls under model development and therefore the scope of GMD. It is my understanding that validation/ model evaluation manuscripts are also permissible in GMD, but I find the model validation part somewhat lacking (see next point)

**Response:**

Please refer our response on the item (3) above. Besides, we conducted an additional experiment to address to your concern. Please refer to our response in the next item.

**(25) Validation/ comparison to other methods**

The authors test the CNN predictions against true biomes, which produces an approximate 50% success rate. The authors also address the limitations of the model, such as the fact that biome change are transient and that real world biomes are much more fractured compared to modeled biomes due to human managements and climate conditions suitable to several biomes/ or plant functional types. Apart from this 1-1 comparison, there is no additional validation against other methods. The authors outline in the introduction several methods for predicting biomes (either empirically or based on pyhsiological limits of vegetation), but never address how their method compares to methods of less, similar or higher complexity. For example, does their method outperform the HLZ scheme or what else is being gained by throwing machine learning at this problem? I want to be clear here: I am not saying that this is not a valid and useful approach, but I don't think that the authors provide sufficient discussion to establish this.

**Response:**

We conducted an additional experiment for comparing the accuracy of PNV map reconstruction between the HLZ scheme and our method. Other PNV mapping schemes introduced in our manuscript cannot be compared directly with our method because they require additional data such as soil physics and topography. The scheme of Woodward & Williams (1987) is the only exception, but it gives multiple vegetation types for a given climatic condition, and hence it cannot be compared with our method directly. Following sentences, new tables S11, and new figure S5, were added to the revised manuscript.

Following sentences were inserted in the Methods section (L171):

--Finally, we conducted an additional experiment for comparing the accuracy of PNV map reconstruction between the HLZ scheme and our method using common training data set. We developed a look-up table of the most common PNV for each combination of annual precipitation class and annual mean bio-temperature class, consistent with the HLZ scheme. Note that the HLZ scheme employs a hexagon table, but we employed a cross-tabulation table for simplicity. CRU annual climate and ISLSCP2 PNV map were used for generating the table. Then the table was applied to all climatic datasets we employed in this study, drawing reconstructed PNV maps for comparison.

Following sentences was inserted in the Result & Discussion section (L235): --The accuracy of PNV reconstruction using the HLZ look-up table for each climate data set is 50.0% for CRU, 43.2% for NCEP/NCAR, 44.71% for HadGEM2-ES, and 37.2% for MIROC-ESM. These values are lower than any of our models trained with annual precipitation and annual mean bio-temperature (all models of Table S2 and model 5 in Table S3). This comparison shows that our method delivers a more accurate reconstruction of the PNV map even if seasonality was not taken into consideration. Consistent with the biome map from the CNN model trained by annual climate (Fig. 1b), the look-up table of the most common PNV (Table S11) lacks tropical deciduous forest and temperate broadleaf evergreen forest. Besides, the look-up table also lacks temperate needle-leaf forest and boreal deciduous forest. Probably, the coarse resolution of the look-up table cannot provide a climate range where these vegetations become the most common vegetation type.

Following figure and its caption were added as new Figure S5.

-- Biome map generated by the look-up table of the most common PNV for each combination of annual precipitation class and annual mean bio-temperature class, consistent with the HLZ scheme (Table S11). Historical CRU annual climate and ISLSCP2 PNV map were used for generating the table. Then the table was applied to historical data of the (a) CRU, (b) NCEP/NCAR reanalysis, (c) Had2GEM-ES data, and (d) MIROC-ESM.

Following table and caption were added as new Table S11.

-- Most common PNV type and its probability (in parenthesis) for each combination of eight annual precipitation classes and six bio-temperatures classes. For cells with less than five grids are indicated as "NA." Definitions of PNV type numbers are as follows.

- 1, Tropical Evergreen Forest/Woodland
- 2, Tropical Deciduous Forest/Woodland
- 3, Temperate Broadleaf Evergreen Forest/ Woodland
- 4, Temperate Needleleaf Evergreen Forest/Woodland
- 5, Temperate Deciduous Forest/Woodland
- 6, Boreal Evergreen Forest/Woodland
- 7, Boreal Deciduous Forest/Woodland
- 8, Evergreen/Deciduous Mixed Forest
- 9, Savanna
- 10, Grassland/Steppe
- 11, Dense Shrubland
- 12, Open Shrubland
- 13, Tundra
- 14, Desert

**15, Polar Desert/Rock/Ice**

|       | Precipitation class (mm/yr) |             |             |             |             |       |       |       |  |
|-------|-----------------------------|-------------|-------------|-------------|-------------|-------|-------|-------|--|
|       | $62.5 \sim$                 | $125\sim$   | 250~        | 500~        | 1000~       | 2000~ | 4000~ | 8000~ |  |
| 0.75~ | 13 (66.3 %)                 | 13 (67.9 %) | 13 (62.7 %) | 13 (47.7 %) | 15 (85.7 %) | NA    | NA    | NA    |  |

| Bio-temperature class
(Celsius) | 1.5~    | 13 (36.5 %) | 8 (49.4 %)  | 13 (47.5 %) | 8 (43.4 %)  | 13 (55.2 %) | NA         | NA         | NA         |    |
|------------------------------------|---------|-------------|-------------|-------------|-------------|-------------|------------|------------|------------|----|
|                                    | (*      | 3.0~        | 14 (65.2 %) | 8 (35.1 %)  | 8 (43.0 %)  | 6 (46.1 %)  | 6 (57.6 %) | 6 (58.8 %) | NA         | NA |
|                                    | Celsius | 6.0~        | 14 (60.9 %) | 10 (58.0 %) | 10 (56.4 %) | 5 (27.1 %)  | 5 (43.1 %) | 6 (42.5 %) | NA         | NA |
|                                    | 9       | 12.0~       | 14 (86.7 %) | 12 (52.2 %) | 12 (42.8 %) | 9 (41.0 %)  | 9 (23.8 %) | 1 (72.9 %) | 1 (59.1%)  | NA |
|                                    |         | 24.0~       | 14 (93.5 %) | 14 (48.5 %) | 11 (25.9 %) | 9 (47.8 %)  | 1 (41.1 %) | 1 (93.3 %) | 1 (96.2 %) | NA |

**(26) Implementation of CNN**

Based on the supplementary information, the CNN (LeNET) is run with default parameters and input parameters are air temperature and precipitation visualized as RGB images, which each image encoding a log transformed and normalized value for the two variables as color. The authors also conduct several experiments (see supplementary tables), but overall all of these have almost equal performance. I am wondering in this context, why this is the case. Is this something that has to do with the CNN that could be overcome by changes to model training/ model architecture changes or has this to do with the fact that the CNN is already extracting all the information that is extractable from the training data.

I feel that this may be the case, considering that CNNs are conventionally used to classify images/photos that are very complex (such as is this a dog or a cat), while the images fed into the CNN are very simple monocolor images. Once again this is an open question that could be addressed in additional discussion.

Specific comments

**Response:**

LeNet is the first CNN, and it was originally developed for classifying handwritten digits (i.e., ten categories). Still, LeNet seems to have sufficient ability to extract most of all the information contained in our training data irrespective of how it is visualized. For adding this point of view, we inserted the following sentences into the last paragraph of the discussion (L274.)

-- We compared performances of models trained by four different types of VCE representation of annual precipitation and average annual bio-temperature, and all models have an almost equal performance (Table S2). This result might indicate that LeNet perfectly extracts at least two variables irrespective of how visualized.

(27) Introduction: I am missing some information about what motivates this model application and why predicting future biomes using AI may be useful.

**Response:**

For clarifying our motivations of this study, we replaced the last paragraph of the introduction with following sentences (L65). In addition, the sentence in L45-46 in the original manuscript was removed, as it's a duplicated description of our research purpose.

-- Using a CNN approach, we demonstrate an accurate and practical method to construct empirical models for operational global biome mapping. To the best of our knowledge, this is the first application of CNN to reconstruct a global biome map. We only employed a small number of climatic variables for input to examine how CNN improves the reconstruction accuracy compared to the classical HLZ scheme. We follow Ise and Oba (2019) and Ise and Oba (2020), a vital option for training CNN with a small number of input variables. This method represents climatic conditions using graphical images and employs them as training data for CNN models. After evaluating the accuracy of the biome map reconstructed by this method, we applied the trained CNN to climatic scenarios toward the end of the 21st century to demonstrate a possible model's application to predict the shift in the global biome map under changing climate.

(28) L72: ISLSCP2: Given that ISLSCP2 is potential land cover for the training, it would be good to discuss any potential issues with this dataset. Is this an unbiased representation of the true potential land cover.

**Response:**

We inserted the following sentence that explains the nature of this data set (L78): -- The ISLSCP2 dataset represents the world's vegetation cover that would most likely exist now in equilibrium with present-day climate and natural disturbance in the absence of human activities.

We also prepared additional explanations for the ISLSCP2 dataset, but we think it is too much for this manuscript. If you think it's better to add these sentences on the manuscript, please let us know via editor, then we will do so!

-- This PNV data was delivered using the global 1km land cover classification data set of Loveland et al. (2000). The most dominant "remnant" land cover type for each grid box was assigned as the PNV type. For grid boxes dominated by land use, simulation output of a process-based vegetation model (Haxeltine and Prentice, 1996) was employed to fill the PNV. (29) L103: "mean of positive air temperature" > I am a bit confused about the positive. how are negative air temperatures treated? I would also encourage to replace positive with 'above freezing' for clarity.

**Response:**

Negative air temperatures were treated as zero for calculating the bio-temperature. According to your suggestion, we replaced the "positive" with "above freezing" in the definition of the bio-temperature. Besides, we found a mistake in the definition of the bio-temperature: It was calculated based on monthly air temperature, not daily air temperature, as was explained in our previous manuscript. Therefore, we changed your mentioned phrases as follows.

Previous sentence (L101 of the previous manuscript):

-- Here, bio-temperature was defined as the mean of positive daily air temperatures.

New sentence (L107):

-- Here, bio-temperature was defined as the mean of above freezing monthly air temperatures.

Previous phrase (L24 of the previous manuscript): -- mean of positive air temperature

New phrase (L26):

-- mean of above-freezing air temperature

(30) L113: "the model with monthly mean air temperature and monthly precipitation had the highest test accuracy"

> given that biomes are most often visualized along air temperature and precipitation axes, this does not seem to be surprising. Humidity and SW radiation may somewhat covary with T and P. I am wondering given that the CNN allows for 3 channels, whether there is some other variable (either climate or altitude) that may be useful to add.

**Response:**

Right. Adding variables (other than humidity and short wave radiation) would improve the vegetation map reconstruction. Especially, adding a topographical variable would be pretty helpful at the sub-grid scale. I inserted the following sentence at the discussion part (L310):

-- Another possible extension is simply adding one more variable that tightly controls

PNV at sub-grid scales (such as altitude, slopeness, or slope aspect) into the VCE because one of the three RGB channels is empty in our model.

(31) Section 2.3: Training of the monthly CNN. The authors should elaborate here on the procedure for using monthly data.

**Response:**

For guiding readers, we added the following phrase at the end of section 2.3 (L140). -- The annual and monthly climate training procedures are identical except for its VCEs.

(32) L190-194: I am not fully following this reasoning which seems to completly discout allocation disagreement. What the authors say may be true, but I don't think this is proven based on the information provided in the manuscript. One problem with this may be the map representation of results, which makes in depth comparisons and deep dive into potential reasons for model misses difficult.

**Response:**

Honestly, we cannot understand the point here. We feel simple map comparisons are not enough, so we calculated the quantity disagreement and allocation disagreement, which (we believe) make in-depth comparisons and deep dive into potential reasons for model failures.

(33) L195: "Table S9 compared the dependence of reconstruction accuracy on combinations of climate datasets for training and test climate datasets"
> I am a bit confused by this, given that the authors reasoned that using the same dataset for train and test could lead to overfitting and then argue here that using the same dataset for train and fit leads to higher accuracies which show robustness of the approach.

**Response**:**

We agree. We changed the following phrase, which interprets the result of this experiment.

Previous phrases (L197 of the previous manuscript):

-- These results suggest that uncertainty in historical climate reconstruction is a larger source of failure in reconstructing biome distribution than the dependency of training on a particular climate dataset.

New phrase:

-- These results suggest that uncertainty in historical climate reconstruction and overfitting are more significant sources of failure in reconstructing biome distribution than the dependency of training on a particular climate dataset.

(34) L282: "Since this method is simply an application of image classification AI, it demands much less technical skill and computer resources compared to other modern techniques such as those evaluated by Levavasseur et al. (2012), Levavasseur et al. (2013), and Hengl et al. (2018), for example."

> I am not sure that this is a fair comparison. One could similarly run a versy simple logistic regression or ANN from a standard package such as scikitlearn, which can easily be executed on a standard desktop PC.

**Response:**

We agree. We replaced your mentioned sentence as follows (L325).

-- Since this method is simply an application of image classification AI, it does not demand much technical skill and computer resources.

**Correction of Erratum**

We realized that the caption of figure 4 was a duplication of that of figure 3. Therefore, taking the opportunity of this revision, we replaced it with the following correct one (L490).

-- Predicted biome maps under climatic scenarios from 2091 to 2100. Monthly means of four sets of forecasted climatic conditions derived from combinations of two climate models (i.e., Had2GEM-ES and MIROC-ESM) and two RCP scenarios (i.e., RCP2.6 and RCP8.5). These means were applied to the CNN model that was trained by the current biome distribution map, as well as the present climatic condition derived from the CRU dataset. Color definitions are available in Figure 1.

---

## Author Response (AR2)

Dear topical editor,

We thank the referees for their constructive comments, and we appreciate your help and patience. We have revised the manuscript according to the referee's comments and suggestions. Below is our one-to-one response to your concerns. Throughout this letter, given words are written in blue and numbered consecutively.

Kind regards, Hisashi SATO on behalf of all authors

**Referee #1**

(1) The authors have addressed all my comments from the discussion paper. I believe there is one new sentence that needs to be corrected (line 276 following), other than that the manuscript can be accepted as is:

For quantifying methodological uncertainty might also result from comparison of performances between correlative and process-based models in 'unsuitable' outside the environmental space of the training data (Yates et al., 2009).

**Response:**

We greatly appreciate your positive evaluation for our manuscript! For your mentioned sentence, we changed as followings (L277).

-- For quantifying methodological uncertainty might also result from comparing performances between correlative and process-based models in 'unsuitable' outside the environmental space of the training data (Yates et al., 2009).

**Referee #2**

(2) The authors have provide responses to most of my comments and I believe that the manuscript is improved. I am still somewhat reserved about the validation part of the model (see below) but acknowledge that this is the first application of CNN to biome classification which makes this novel and therefore a valid contribution to scientific discourse.

**Major comments:**

Regarding the HLZ model validation: I am grateful for the additional model comparison. I think that this is crucial. I also believe that a comparison between HLZ and CNN should be part of the main body of the paper (i.e. a figure or a table) rather than just providing supplementary data for the HLZ model. Similarly one could ask what were to happen if the HLZ would be run more finegrained. I am assuming that changing the bin sizes for the HLZ would potentially improve things until smaller bin sizes would not longer have an effect (e.g. convergence to a best possible model). I just want to make sure that the comparison between the CNN and the HLZ is as fair as possible.

**Response:**

We greatly appreciate your positive evaluation of our manuscript! As you suggested, we conducted additional experiments of the HLZ scheme using finergrained bin sizes. Its results are presented as table 2 in the main-body of this manuscript.

For describing the new experiment, we inserted following phrases in the L175. -- The bin sizes of the HLZ scheme are six for the annual mean bio-temperature class and eight for the annual precipitation class. As these coarse-grained bin-sizes would potentially depress the accuracy of the PNV simulation, we also developed look-up tables of  $12 \times 16$ ,  $24 \times 32$ , and  $48 \times 64$  bin-sizes to ensure that the comparison between our model and the HLZ scheme is as fair as possible.

With this new experiment, the corresponding paragraph (L235-242 of the previous manuscript) was replaced by the following paragraph (L240-243).

-- Accuracies of PNV reconstructions using the HLZ look-up tables for each climate data-set increase with the resolution of bin sizes for climate classifications (Table 2). It reaches quasi-equilibrium at 24 bio-temperature classes × 32 precipitation classes, which delivers nearly identical results with the CNN model. This result demonstrates that our VCE method extracts the best possible distribution of the most plausible PNV in a two-dimensional space of climatic variables. A new table, which explain the experiment, was inserted.

Table 2 CNN model and HLZ models accuracies for biome distribution simulations. CNN model corresponds to the top row model of table S2 (a RGB colour tile). Four HLZ models have different bin sizes for climate classifications (bio-temperature class × precipitation class). Each model was trained with the CRU dataset and adapted to the all-climate datasets (i.e., agreements of the CRU dataset correspond to the training accuracy, while other climate data correspond to the test accuracy). In addition, for each climate dataset, the annual mean bio-temperature and annual precipitation from 1971 to 1980 were log-transformed before use.

| CNN model | HLZ models                                        |                                                                                                                                                       |                                                                                                                                                                                                                                                                 |                                                                                                                                                                                                                                                                                                                                |
|-----------|---------------------------------------------------|-------------------------------------------------------------------------------------------------------------------------------------------------------|-----------------------------------------------------------------------------------------------------------------------------------------------------------------------------------------------------------------------------------------------------------------|--------------------------------------------------------------------------------------------------------------------------------------------------------------------------------------------------------------------------------------------------------------------------------------------------------------------------------|
|           | 6×8                                               | 12×16                                                                                                                                                 | 24×32                                                                                                                                                                                                                                                           | 48×64                                                                                                                                                                                                                                                                                                                          |
| 58.3 %    | 50.0 %                                            | 54.9 %                                                                                                                                                | 58.1 %                                                                                                                                                                                                                                                          | 60.4%                                                                                                                                                                                                                                                                                                                          |
| 45.6 %    | 43.2 %                                            | 45.8 %                                                                                                                                                | 44.9 %                                                                                                                                                                                                                                                          | 44.0%                                                                                                                                                                                                                                                                                                                          |
| 48.6 %    | 44.7 %                                            | 46.8 %                                                                                                                                                | 48.2 %                                                                                                                                                                                                                                                          | 46.9%                                                                                                                                                                                                                                                                                                                          |
| 41.3 %    | 37.2 %                                            | 40.1 %                                                                                                                                                | 40.8 %                                                                                                                                                                                                                                                          | 39.5%                                                                                                                                                                                                                                                                                                                          |
|           | CNN model
58.3 %
45.6 %
48.6 %
41.3 % | CNN model         6×8           58.3 %         50.0 %           45.6 %         43.2 %           48.6 %         44.7 %           41.3 %         37.2 % | HLZ n           CNN model         HLZ n           6×8         12×16           58.3 %         50.0 %         54.9 %           45.6 %         43.2 %         45.8 %           48.6 %         44.7 %         46.8 %           41.3 %         37.2 %         40.1 % | HLZ wodels           CNN model         6×8         12×16         24×32           58.3 %         50.0 %         54.9 %         58.1 %           45.6 %         43.2 %         45.8 %         44.9 %           48.6 %         44.7 %         46.8 %         48.2 %           41.3 %         37.2 %         40.1 %         40.8 % |

With these modifications, we deleted table S11 and Fig. S5 from the previous manuscript, as they are no longer needed very much. We renumbered figure S6-S9 on the previous manuscript as figure S5-S8.

(3) Regarding my original point on the ability of the CNN (response 26): I am still not sure to what extent using a CNN is really the appropriate tool with respect to complexity.

In my mind the complexity of approach would be:

HZL > Logistic Regression > ANN > CNN

I therefore have no problem acknowledging that the CNN is capable of extracting all information from the images. As the authors say the CNN was developed to classify handwritten numbers and the benefit of the CNN is that the CNN is capable of using image information to extract complex features from images (in the number example this would for example be shadings/ edges/ vertical and horizontally oriented lines/ curves). In the case of the biome classification: there are exactly two features that are prescribed. T and P information as two-channel pseudocolor. The exactly same information could be presented to a logistic regression or an ANN as numeric information (without having to first convert numeric information into images, which is - while straightforward in nature - laborious.

An ANN implemented in any ML package (sklearn, tensorflow, ...) or logistic regression in sklearn or any other package and an ANN in Keras or tensorflow could be implemented fairly easily. Given the limited amount of data and variables, I don't think that training such an ANN would take long. I have trained similarly sized ANN on current generation PCs.

Given that simple models allow for better explainability of results, I am really wondering whether CNN is the right tool. Therefore, it would have been nice to see how the CNN compares with for example logistic regression, given that logistic regression is also data driven, would make use of all the data (but remains a linear model). ANN on the other size is capable of extracting non-linear information. So an interesting question would be to see whether adding non-linearity adds benefit to the classification. Beyond that I am doubtful that the CNN would improve classification over the ANN, since the ANN should theoretically be capable of extracting all the information from T an P channels.

I know that this is not the manuscript the authors have written.

In that light, I don't think that the authors manuscript has any methodological problems and could be published subject to minor revisions.

**Response:**

If seasonal patterns are not considered for simulating biome maps (such as HLZ), we agree with you. All ANNs (Artificial Neural Networks) that can treat non-linearity would deliver the same result.

However, using our method, the seasonal pattern of multiple climatic variables can be employed for machine learning without any indexical expression. This is the most significant advantage of our method because indexical expression reduces the amount of information and adds a source of arbitrary. Among image classification ANNs, LeNet CNN has the lowest complexity, so we believe LeNet CNN is the right tool for this issue. To clarify the above advantage of our method, the introduction section was modified as follows.

Following paragraph in the L70-72 of previous manuscript was moved to L62 in the new manuscript.

--After evaluating the accuracy of the biome map reconstructed by this method, we

applied the trained CNN to climatic scenarios toward the end of the 21st century to demonstrate a possible model's application to predict the shift in the global biome map under changing climate.

The sentences in L54-58 in the previous manuscript were moved to the L68 in the new manuscript, with some adjustments of conjunctions as follows.

-- To account for seasonal variability, previous correlative climate-vegetation models needed to pre-define representative variables. For example, Levavasseur et al. (2013) divided each climatic variable into four "seasonal" predictors by averaging data corresponding 3-month periods (i.e., DJF for winter, MAM for spring, JJA for summer, and SON for fall). By contrast, the method we employed can automatically extract nonlinear seasonal patterns for climatic variables that are relevant in biome classification.

The following new sentence was inserted in L73.

-- In other words, it enables CNNs to learn the seasonal pattern of multiple climatic variables without any indexical expression, which would reduce the amount of information and add a source of arbitrary.

(4) Response 32: I am referring here to the visualization of results. The authors chose maps and summary statistics (accuracy) as their primary tool to convey information and I feel that for example looking at which biomes are more likely subject t allocation disagreement may be another way of better understanding why the model produces the results it produces, which for data driven methods is important but less straight forward.

**Response:**

We inserted the following figure to directly compare the biome compositions between maps. This figure is referred at the first paragraph of section 3.1. With this new figure 2, we renumbered figures 2-4 on the previous manuscript as figures 3-5.

**Figure 2**

Global biome compositions of the observation-based map (a) and simulated maps from CNN models trained by monthly mean climate (b) and annual mean climate (c) of CRU climate data spanning of 1971 to 1980. These CNN models were adapted to the four climatic datasets (CRU, NCEP, Had2GEM-ES, and MIROC-ESM) spanning the same period of the training data.

(5) L74: "We follow Ise and Oba (2019) and Ise and Oba (2020), a vital option for training CNN with a small number of input variables." > I suggest neutral language:
"We follow the method of Ise and Oba (2019) and Ise and Oba (2020) for training ... "

**Response:**

We agree. The mentioned phrase was changed in accordance to your suggestion (L67).

(6) On a side note to comment 30: Including elevation may add additional information for the model to make use of (our ANN model in: Gerken, T., Ruddell, B.L., Yu, R. et al. Robust observations of land-to-atmosphere feedbacks using the information flows of FLUXNET. npj Clim Atmos Sci 2, 37 (2019). https://doi.org/10.1038/s41612-019-0094-4) did include for example elevation.

**Response:**

We cited the Gerken et al. (2019) as an example for how including elevation improve the model performance (L313).

-- For example, for geographically extrapolating flux data observed at flux tower sites, Gerken et al. (2019) trained artificial neural networks (ANN) using the elevation of each tower site.